# Solving Packing Problems by Conditional Query Learning

**Dongda Li[1], Changwei Ren[2], Zhaoquan Gu[1], Yuexuan Wang[2], Francis C.M. Lau[3]**
[1]Cyberspace Institute of Advanced Technology, Guangzhou University, Guangzhou, China
[2]College of Computer Science, Zhejiang University, Zhejiang, China
[3]Department of Computer Science, The University of Hong Kong, Hong Kong, China
`lidongda@gzhu.edu.cn, rcw@zju.edu.cn, zqgu@gzhu.edu.cn`
`amywang@zju.edu.cn, fcmlau@cs.hku.hk`

## Abstract

Neural Combinatorial Optimization (NCO) has shown the potential to solve traditional NP-hard problems recently. Previous studies have shown that NCO outperforms heuristic algorithms in many combinatorial optimization problems such as the routing problems. However, it is less efficient for more complicated problems such as packing, one type of optimization problem that faces mutual conditioned action space. In this paper, we propose a Conditional Query Learning (CQL) method to handle the packing problem for both 2D and 3D settings. By embedding previous actions as a conditional query to the attention model, we design a fully end-to-end model and train it for 2D and 3D packing via reinforcement learning respectively. Through extensive experiments, the results show that our method could achieve lower bin gap ratio and variance for both 2D and 3D packing. Our model improves 7.2% space utilization ratio compared with genetic algorithm for 3D packing (30 boxes case), and reduces more than 10% bin gap ratio in almost every case compared with extant learning approaches. In addition, our model shows great scalability to packing box number. Furthermore, we provide a general test environment of 2D and 3D packing for learning algorithms. All source code of the model and the test environment is released.

## 1 Introduction

How to pack boxes with the smallest bin size? With the development of globalization and E-Commerce, this question becomes more and more important. Boxes are packed to various bins such as shipping container and boxcar. Many of these boxes are made by paper or plastic; packing boxes in a more efficient way can greatly reduce material cost or shipping energy.

The Bin Packing Problem (BPP) is one of the classic integer combinatorial optimization problems and it has been extensively studied for decades (Wu et al., 2010). It is shown that BPP is a strongly NP-hard problem (Martello et al., 2000), which requires exponential time to generate the optimal solution. Some heuristic algorithms (Baltacioglu et al., 2006) try to obtain a nearly optimal solution within polynomial time, but these methods require explicit rules for every specific problem setting. When the setting changes even very slightly, the original method cannot work properly. On contrast, the explicit or hand-crafted rules can be interpreted as policies by neural networks to make decisions, which are insensitive to problem settings. Neural networks have achieved great success in many domains, such as computer vision (Liu et al., 2017), natural language processing (Vaswani et al., 2017), speech recognition (Chan et al., 2016), etc. Inspired by these booming techniques, many studies (Bello et al., 2016) adopt the neural networks and the recent advances in artificial intelligence to solve the classic combinatorial optimization problems, such as the Travelling Salesman Problem (TSP), the Vehicle Routing Problem (VRP), etc. Some latest works propose the pointer network (Vinyals et al., 2015b) and utilize the attention mechanism with reinforcement learning (Nazari et al., 2018; Kool et al., 2019) to solve the TSP and the routing problems respectively.

There are also some learning-based attempts for the packing problem, which utilize reinforcement learning in the neural network model since the optimal solution is unknown beforehand. For exam-

ple, MTSL (Duan et al., 2019) separates selecting and rotating step by selected learning, and applies the transitional greedy method to perform final positioning step. Laterre et al. (2018) enlighten by AlphaZero (Silver et al., 2017) adopt Monte Carlo Tree Search (MCTS) with self-competition reinforcement learning to solve the packing problem, but it is restrained to pack boxes that have been preliminarily divided from a bin. Cai et al. (2019) simply use reinforcement learning to get some packing results, which serve as the initialization to accelerate the original heuristic algorithms.

However, these approaches either miss the connection between sub-actions or combine handicraft rules with a learning algorithm. Without the sub-action connection, the learning process becomes partial observable Markov Decision Process (MDP) (Sutton & Barto, 2018), which is hard to generalize a good policy for the lack of information. Some methods generate all possible sub-action sequences at the same time, it is still non-trivial for a neural network model to produce many mutual related outputs in a single forward prorogation even though the setting of these methods are strict MDP. These methods combining handicraft rules are not only difficult to achieve an optimal solution, but also sensitive to problem settings.

To the best of our knowledge, there is no end-to-end learning algorithm that solves standard packing problems. In this paper, we propose a Conditional Query Learning (CQL) model that directly addresses the gap between sub-actions. With the inner conditional query mechanism, the learning process becomes fully observable MDP, which makes the problem easier to apply the reinforcement learning. Compared with the model that outputs several sub-actions simultaneously, CQL model has smaller action space per forward step. Benefit from the small action space feature, we can apply a simpler model to fit the policy, which work more efficiently. In addition, it would not sacrifice performance with the small action space. As a result, we do not require handicraft rules to feed the gaps between sub-actions since they are provided by conditional queries.

Specifically, the packing problem requires three mutual conditioned sub-actions: box selecting, rotating and positioning. To fill the gap between sub-actions, we adopt the CQL model as follows. First of all, the packing problem is formulated as an MDP to apply reinforcement learning. Then the previous sub-actions are embedded as a query to the next sub-action decoder. After all three sub-actions are generated, the environment performs one packing step and updates the observation. Finally, we adopt the actor-critic algorithm to update the model parameters.

We conduct extensive experiments to evaluate the models and the results show that the CQL model greatly outperforms the vanilla model which produces sub-actions in one step forward propagation without query. In addition, the CQL model achieves lower bin gap ratio in both 2D and 3D packing compared with extant learning approaches and heuristic algorithms. Specifically, our model improves 7.2% space utilization ratio in 3D packing (30 boxes) compared with genetic algorithm, and reduces more than 10% bin gap ratio in almost every case compared with the state-of-the-art learning approaches. Furthermore, numerical results show that CQL greatly outperforms other methods when the scale of the problem increases. The learning carve and the variance of results also illustrate that CQL makes the training process more stable.

The contributions of this paper are summarized as follows:

- We propose the first end-to-end learning algorithm that solves the standard packing problem for both 2D and 3D settings;
- We propose the conditional query learning (CQL) model to address the packing problem that has mutual conditioned multi-dimension actions;
- We combine the conditional query learning with an attention mechanism to construct a new learning framework;
- We conduct extensive experiments and the results show that our model outperforms both heuristic algorithms and the state-of-the-art learning-based models.

We also release our model implementation and 2D&3D packing environment for the community to test their algorithms.

The rest of the paper is organized as follows. We introduce related works in the next section. We introduce the preliminaries in Section 3. Design of CQL model is presented in Section 4, We implement CQL model and illustrate the comparison results in Section 5. Finally, we conclude this paper in section 6.

## 2 RELATED WORKS

Getting an optimal solution of combinatorial optimization problems are computationally heavy, optimal labeled data for supervised learning is expensive. Hence, when using Neural Networks (NNs) to solve these problems, one solution is to apply heuristic algorithm results as labeled data, but the performance of this approach cannot be better than the performance of the heuristic algorithm. The other solution is to apply reinforcement learning that makes the algorithm learn from its own experience, which is possible to produce better results than existing algorithms. Here we focus on the reinforcement learning approaches and introduce some related works of reinforcement learning and neural combinatorial optimization.

### 2.1 SEQUENCE PROBLEM IN NEURAL COMBINATORIAL OPTIMIZATION

Enlighten by the recent success of Neural Networks (NNs), especially the fast progress in sequence to sequence model, which is initially used in Neural Machine Translation (NMT) (Bahdanau et al., 2014; Vinyals et al., 2015a; Luong et al., 2015; Vaswani et al., 2017; Shen et al., 2018). Because many combinatorial optimization problems have similar input and output structure as NMT, many studies adopt NNs to solve sequential combinatorial optimization problems. Vinyals et al. (2015b) propose Pointer Networks, which uses attention as a pointer to select a member of the input sequence as the output. Bello et al. (2016) and (Nazari et al., 2018) view TSP and VRP as MDP respectively, and they both apply policy gradient algorithm to train their models. Kool et al. (2019) further improve the result of routing problem using attention model (Vaswani et al., 2017).

### 2.2 REINFORCEMENT LEARNING WITH PARAMETERIZED ACTION SPACE

Different from routing problems that only require one object selected from input sequence per step, the packing problem requires three sub-actions to pack a box into the bin. This kind of action space is called parameterized action space (Masson et al., 2016) in reinforcement learning, which requires the agent to select multiple types of action every action step. Hausknecht & Stone (2015) expand DDPG to parameterized action space and test it on RoboCup soccer game. But this approach suffers from $tanh$ saturation problem in continuous action space, so they apply inverse gradients trick to address it. Masson et al. (2016) introduce Q-PAMDP algorithm, which alternates learning action selection and parameter selection polices to make the training process more stable, but the parameter policy have to output all the parameters for all discrete actions. The output size of the parameter policy can be explosion when the problem has large high dimensional parameters with large discrete actions. The authors of Wei et al. (2018) propose a hierarchical approach, they condition the parameter policy on the output of discrete action policy, and they apply Variational Auto-Encoders (VAE) trick to make the model differentiable.

However, all these researches only divide action to two parts, namely, a discrete action and a set of actions as parameters that may include discrete or continuous actions. But the problems like packing contains several mutual conditioned actions that can not directly view as conventional parameterized action space problem, otherwise the number of outputs of the second model will be the product of the candidate output of each sub-actions, which makes the model has too many output options and is hard to generalize the problem and learn to produce a good solution.

### 2.3 PACKING PROBLEM

As mentioned before, the packing problems are such problem that have mutually conditioned sub-actions, and there are some works trying to solve this problem by NCO. Enlighten by AlphaZero Silver et al. (2017), Laterre et al. (2018) apply MCTS self play to search the better solution and learn how to pack, but their method only applies to the dataset which boxes are divided by random cuts from a regular bin. Duan et al. (2019) propose a selected learning approach solving 3D flexible bin packing problem which balances sequence and orientation complexity. They adopt pointer networks with reinforcement learning to get the box selection and rotation results, and apply greedy algorithm to select the location of boxes, which is not end-to-end learning method and can not get better than greedy algorithm. More importantly, this hybrid method do not view the entire packing problem as one complete optimization process, so the learning process only tries to optimize part of problem. At the same time, different algorithms of sub-actions may have conflict goal in optimization process.

Different from previous methods, we simply embed the previous actions as an attention query for the model to reason the subsequent actions and after the full action step is finished, we perform one-step optimization along every sub-action model. In this way, we reduce the total output size to the sum of all sub-action candidate output, which makes the model smaller and easier to learn.

## 3 PRELIMINARIES

In this section, we introduce the Bin Packing Problem (BPP) and formulate it as a MDP.

### 3.1 BIN PACKING PROBLEM

Bin packing problems are a class of optimization problems in mathematics that involve attempting to pack objects together into containers. The goal is to either pack a single bin as densely as possible or pack all objects using as few bins as possible. For simplicity, we adopt the first goal, in which we have a number of boxes and we want to use minimal size of bin to pack all these boxes. Specifically, we have fixed bottom size rectangle (2D) or cube shape (3D) bin, and the object is to minimize the final height of bin and achieve higher space utilization ratio. The problem is the strip packing problem with rotations (Christensen et al., 2017), which is a subclass of BPPs, and we put the formal definition of this problem in Appendix 7.1. In the following sections, we elaborate our approach in the 3D BPP if not specifically stated.

In respect of NCO, the packing procedure of each box can be divided into three sub-actions:

1. Selecting target box from all unpacked boxes.
2. Choosing the rotation of the selected box.
3. Outputting coordinates relative to the bin of the rotated box.

These three sub-actions are ordered and mutual conditioned, that is, to select a rotation, choosing the box to be rotated should be done first, and each previous decision serves the following one. Each sub-action can not be viewed as an independent optimization process, otherwise, there may be conflicts between optimization processes, resulting in sub-optimal and even some strange solutions.

### 3.2 FORMULATING PACKING PROBLEM AS MARKOV DECISION PROCESS

Since the packing problem is largely NP-hard, getting an optimal solution in acceptable time is not realistic, thus we have to view the problem as an MDP and adopt reinforcement learning to make the agent learn from experience. In MDP, the probabilities given by $p$ completely characterize the environment's dynamics. That is, the state of the environment must include information about all aspects of past agent-environment interaction that make a difference for the future, and this kind of state is said to have the *Markov property*.

To make the problem satisfy Markov property, we divide our model to encoder and decoder. The encoder state is $S = \{s_1, s_2, ..., s_n\}$, where $s_i = (s_{p,i}, l_i, w_i, h_i, x_i, y_i, z_i)$, and $s_p$ is a boolean variable that indicates whether the box is packed in the bin. $(l_i, w_i, h_i)$ is box length, width and height, and $(x_i, y_i, z_i)$ is the box left-front-bottom coordinate relative to the bin. From the perspective of a box packing step, giving $s$ is enough for MDP, however, if we divide the box packing step to the three sub-actions, then the rotating and positioning step is not strict MDP. The rotating step has to know which box is selected, and the positioning step has to know both the selected box and rotation from previous sub-actions. Therefore, we propose the attention encoder with dynamic state and conditional query decoder which is introduced in the next section. The detail of environment transition is described in Appendix 7.2.

## 4 CONDITIONAL QUERY LEARNING MODEL

In this section, we introduce the CQL model. Our method adds a conditional query to Multi-Head Attention (MHA) (Vaswani et al., 2017) to make the model capable of solving mutual conditioned multi-dimensional optimization problems.

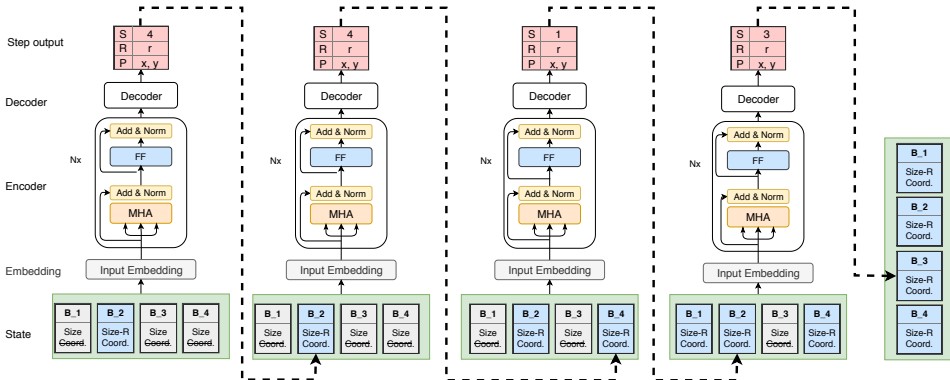

Figure 1: Attention Encoder with Dynamic State for 4 Boxes Case. Every previous packing step output leads to corresponding box state update and re-encodes in the following step.

We divide this section into three parts, namely, encoder, decoder and training. We first show the dynamic state self-attention encoder model.

### 4.1 ATTENTION ENCODER WITH DYNAMIC STATE

As shown in Fig. 1, we adopt Transformer (Vaswani et al., 2017) layers to encode input states. Different from Vaswani et al. (2017), the sequence of boxes are unordered in packing problem, so we remove the positional encoding in Vaswani et al. (2017). The input state of encoder contains box shape, box position, and the boolean variable that indicates whether the box has been packed. In addition, we mask the box position in the input state if the box is not packed.

Unlike machine translation or routing problems which has a fixed input feature in the procedure of inference, in packing problem, the rotation and coordinates of boxes are updated immediately after one box has been packed, which makes the encoding embedding must update after every packing step, rather than keep fixed encoded hidden feature on one training example.

To construct a strict MDP, we design the Attention Encoder with Dynamic State shown in Fig. 1, where we first embed input state to fixed vectors and then feed it to $n$ layers Transformer

Table 1: box size update through rotation

| Rotation | 0 | 1 | 2 | 3 | 4 | 5 |
|---|---|---|---|---|---|---|
| $l',w',h'$ | l,w,h | l,h,w | w,l,h | w,h,l | h,l,w | h,w,l |

that contains a Multi-Head Attention (MHA) layer and FeedForward (FF) layer. Each layer adds a residual connection (He et al., 2016) and batch normalization (Ioffe & Szegedy, 2015). We feed the embedded vector to conditional query decoder introduced later. The decoder chooses the packing box from unpacked boxes and the rotation, coordinates of the packing box. After finishing one step packing, we update the input state of encoder according the decoding result of last packing step, more specifically, set the packed state $s_{p,i}$ to True and update the last packed box shape $(l_i, w_i, h_i)$ according to the box rotation as shown in Table 1 and replace the masked position with the packed box location $(x_i, y_i, z_i)$ as shown in Fig. 1.

### 4.2 CONDITIONAL QUERY DECODER

After encoding the input state to an embedded vector for each box, we design the conditional query mechanism to handle the connection between sub-actions without greatly increasing the complexity of the model. As shown in Fig. 2, for packing problem, the model performs three sub-actions separately, namely, boxes selection, rotation and location for the selected box.

In the box selection step, we first feed the dynamic input state to the encoder and get $N$ hidden vectors $H = (h_1, h_2, ..., h_n)$ as described before. Then we feed it to the selection decoder. In this step, all information from encoder is required, so we use self-attention followed by select head, which consists of several linear layers with $N$ final outputs. To avoid the situation that boxes which have already been packed are selected, we mask the packed boxes on selection output and then perform sampling.

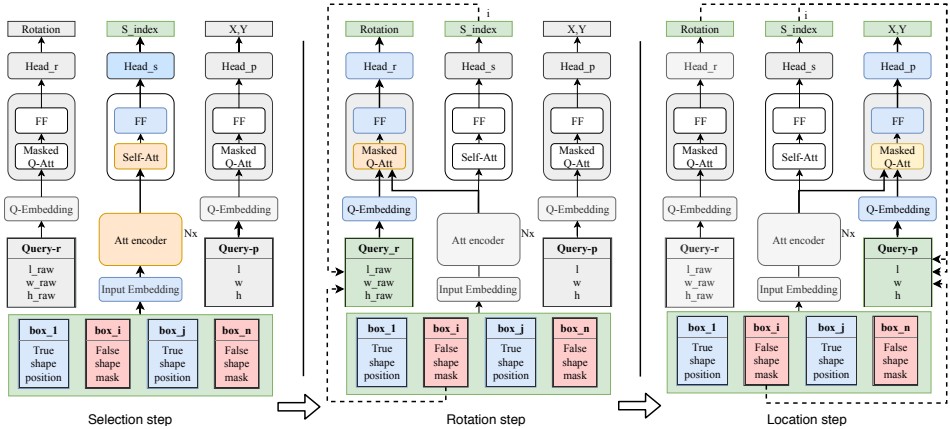

Figure 2: Conditional Query Process. The figure demonstrates the condition query process when packing box $i$, which embed previous sub-action results as conditional query for current decoder.

In the rotation step, we construct a conditional query vector by picking up the selected box shape information from the input state and embedding it by one linear layer. Besides, after choosing the box to pack, the model has to know the boxes which have been put into the bin, so we mask out the attention vector of unpacked boxes as shown in Eq. 2 in scaled dot-product attention 3 of the decoder. At last, we feed the decoder output $h_q$ to rotation head to produce rotation result (2 options for 2D and 6 options for 3D). Where the $W^K$ and $W^V$ are trainable parameters, $k$ and $v$ are key and value vector in attention model, and the $q$ is the embedded query vector.

$$k_i = W^K h_i, \ v_i = W^V h_i \tag{1}$$

$$a_i = \begin{cases} \frac{q^T k_i}{\sqrt{d_i}} & s_{p,i} = True \\ -\infty & s_{p,i} = False \end{cases} \tag{2}$$

$$h_q = \sum_{i=0}^{n} softmax(a_i) v_i \tag{3}$$

In the final step, the model calculates the position of the selected box relative to the bin. To comprise all previous results, the query have to combine rotation of selected box, and we do this by following rules in Table 1 Then it is embedded and fed to the masked MHA. The model may generate coordinates outside the bin, and a fixed bound can not make sure all the boxes do not exceed the bin since the box sizes are different. We bound the packing box coordinates according to the bin width and length after every packing step by moving the boxes outside the bin to the border of the bin.

Throughout the entire forward pass of packing one box, the data stream passes through the encoder and each decoder once. The target query is extracted from the input state based on the previous outputs. By conditional query, the model receives the information from hidden vectors of encoder as well as previous sub-action outputs, which ensures every sub-action decoding step is a strict MDP.

### 4.3 TRAINING

As mentioned earlier, we view the packing process as a MDP and apply reinforcement learning, specifically, the actor-critic algorithm (Konda & Tsitsiklis, 2000). The model we presented earlier is the actor model.

#### 4.3.1 CRITIC NETWORK

The critic network consists of self-attention layers followed by value head. The input of critic is the dynamic state $s$. The graph embedding $\bar{H} = \frac{h_1 + h_2 + ... h_n}{n}$ of the last self-attention layer is fed to value head that contains several linear layers, where $\bar{H}$ is the mean of last hidden vectors of the last attention layer. To make the train process mare stable and easy to tune, we separate the actor network and critic network, that is, the two networks do not share parameters.

### 4.3.2 VALUE FUNCTION ESTIMATION

In the actor-critic algorithm, to accurately estimate the advantages ($\hat{A}_t$) which have a significant impact on optimization performance, a popular technique is learning the state value function $V(s)$ and performing bootstraps to get low variance advantage estimations. However, using only one step bootstrap introduces too much value estimation error. Schulman et al. (2015b) propose the Generalized Advantage Estimation (GAE), which combines n-step bootstrap to produce a more accurate advantage. The GAE balances the bias and variance to achieve stable and accurate advantage estimation. As shown in Eq. 4, where $t$ is the time index in $[0, T]$, with a $T$-steps trajectory segment, and $\lambda$ is the hyper-parameter that adjust the trade-off between bias and variance.

$$\hat{A}_t = -\delta_t + (\gamma\lambda)\delta_{t+1} + ... + (\gamma\lambda)^{T-t+1}\delta_{T-1},$$
$$\text{where} \quad \delta_t = r_t + \gamma V(s_{t+1}) - V(s_t) \tag{4}$$

### 4.3.3 REWARD FUNCTION

Unlike supervised learning, the agent in reinforcement learning studies from experience and tries to find the policy that gets more accumulated rewards. Therefore, how to design the reward signal is a crucial factor for learning process. In the single bin packing problem, the goal is to minimize the height of the bin with given width and length. The most straightforward way is to adopt the negative change of bin height $-\Delta h = h_i - h_{i+1}$ as the reward signal in every packing step. However, this leads to sparse reward which is one of the biggest challenges (Andrychowicz et al., 2017) in reinforcement learning.

$$g_i = WLH_i - \sum_{j=1}^{i}(w_j l_j h_j) \tag{5}$$

$$r = \Delta g_i = g_{i-1} - g_i$$

To address the sparse reward problem, we design the reward signal based on the change of the current volume gap of the bin. As shown in Eq. 5, the volume gap $g_i$ of packing step is defined as current bin volume minus the total volume of packed boxes. Where the $W, L, H$ is the width, length and height of the bin, respectively. The reward of packing step is defined as $\Delta g_i$, then the accumulated reward becomes the final gap of bin, which is linear to the negative final bin height $-H_n$ as formulated in Eq. 6. By doing this, the agent always gets a meaningful reward signal no matter if the total bin height increase in that packing step.

$$R = \sum_{i=1}^{n} r_i = (0 - g_1) + (g_1 - g_2) + ... + (g_{n-1} - g_n)$$
$$= -g_n = -WLH_n + \sum_{j=1}^{n}(w_j l_j h_j) \tag{6}$$

### 4.3.4 TRAINING PROCESS

In every training step, the critic network gets the dynamic state input to estimate the state value, and the actor network performs the three steps described before to get each sub-action output. Thereafter, one-step parameter update is performed for both actor and critic networks. Because our training data is randomly generated and is inexpensive, it is better to use on-policy reinforcement learning which does not has the sample inefficient problem that is shown in off-policy methods (Schulman et al., 2015a).

$$\mathcal{L} = \mathcal{L}_a + \mathcal{L}_c + \beta\mathcal{L}_{ent} \tag{7a}$$

$$\mathcal{L}_{a,\theta_a} = -\hat{A}_t * \sum_{a\in\mathcal{A}} \log[\pi_{\theta_{as}}(s, a_s) + \pi_{\theta_{ar}}(s, a_r) + \pi_{\theta_{ap}}(s, a_p)] \tag{7b}$$

$$\mathcal{L}_c = MSE(V_{\theta_c}(s_t), \hat{A}_t + V_{\theta_c}(s_t)) \tag{7c}$$

$$\mathcal{L}_{ent} = -\sum_{a\in\mathcal{A}}(\pi_{\theta_a} \log \pi_{\theta_a}) \tag{7d}$$

The Eq. 7 formulates the loss function calculation process, which includes actor, critic and entropy loss functions. The policy consists of three sub-action policies as shown in Eq. 7b, The actor loss is the advantage multiply policy gradient that encourage the action which achieves higher accumulate rewards. The critic network is trained through MSE loss 7c. In addition, we add the negative entropy 7d item to loss to avoid the policy is too deterministic and encourage exploration.

---

**Algorithm 1** Conditional Query Learning

1: **Input** Random produce N training set X
2: Initialize actor and critic network parameters $\theta, \phi$
3: **for** t in $\{1,...,N\}$ **do**
4:     Randomly sample a batch from X
5:     Make state $s(s_p, l, w, h, x, y, z)$
6:     **for** j in $0, n/n_{gae}$ **do**               // $n$ is the the number of boxes
7:         **for** k in $0, n_{gae}$ **do**           // Getting a mini-batch for GAE
8:             Get $V_{\theta_c}(s_{j*n_{gae}+k})$        // Critic network for value
9:             $\boldsymbol{H} = \{h_1, h_2...h_n\} = att_{\theta_e}(s_{j*n_{gae}+k})$    // Encoder
10:            Sample box index $i$ from $\pi_{\theta_s}(\boldsymbol{H})$     // Selecting step
11:            Sample box rotation $r$ from $\pi_{\theta_r}(\boldsymbol{H}, q_{r,i})$    // Rotating step
12:            Update $l'_i, w'_i, h'_i$ via Table 1       // Update query based on rotation
13:            Sample position $x, y$ from $\pi_{\theta_p}(\boldsymbol{H}, q_{p,i})$    // Positioning step
14:            Update state $s$ and get reward       // Dynamic state update
15:         **end for**
16:         Calculate advantages by Eq. 4        // GAE
17:         Calculate loss by Eq. 7
18:         $\theta \leftarrow \theta + \alpha_a \Delta \mathcal{L}_a, \phi \leftarrow \phi + \alpha_c \Delta \mathcal{L}_c$
19:     **end for**
20: **end for**
21: **Output** $\theta, \phi$

---

As shown in Alg. 1, after initialization, the algorithm gets $n_{gae}$-step experiences for GAE. In every packing step, the encoder embeds input state to $n * d_{model}$ hidden vectors $\boldsymbol{H}$, and the model gets packing box index, rotation, and position by conditional query process in the decoder (line 10∼line 12). After finished one step packing, the input state and rewards are updated correspondingly.

## 5 EXPERIMENTS

We evaluate the CQL model on 2D and 3D bin packing problems with different packing box numbers and compare the results with heuristic algorithms and learning methods.

### 5.1 PACKING ENVIRONMENT AND DATASET

We design the test environment for 2D and 3D bin packing problems. Here we only describe the 3D condition that can be easily reduced to 2D. In our test environment, $N$ boxes are initialized by random width, length and depth for packing. The bin is initialized by fixed width and length, specifically, from -1 to 1. The agent has to find a solution sequence $S(\boldsymbol{s}, \boldsymbol{o}, \boldsymbol{x}, \boldsymbol{y})$ that uses minimal bin depth and does not have any overlap boxes. The final result is assessed by the accumulated reward which is linear to negative bin height. The environment is designed for reinforcement learning, and we also encourage others to test their algorithms on it.

To fully evaluate the learning model, the test dataset should be hard enough for packing, that is, the variance of box size should not be too large or too small. When the variance is too large, some large boxes will dominate the bin, then other small boxes are more likely to packed below large boxes, which makes the position of small boxes has little impact on the total bin height. When the variance is too small, the boxes are similar to cubes, which makes the problem too easy to solve. Therefore, we choose the dataset with the box sizes sampled from $[0.02, 0.4]$ for 2D and $[0.2, 0.8]$ for 3D after a few tests.

## 5.2 Implementation Details

Both actor and critic networks of our model have two self-attention layers. The box selection decoder has one self-attention layer, and both the box rotation and location decoder have one conditional query attention layer. Each attention layer applies batch normalization. We set the attention hidden dim as 64 both in 2D and 3D cases. Every decoder head has three linear layers with Relu (Nair & Hinton, 2010) activation function. The decoder calculates the probability distribution for each sub-actions. The box selection, rotation and position head generate $N$ box selection choices, 6 rotation choices, and 128 position discrete choices for each axis, respectively. The linear decay learning rate scheduler is applied with Adam (Kingma & Ba, 2014) optimizer and 5e-5, 3e-4 initial learning rate for actor and critic, respectively. We train our model on a single Tesla V100 GPU. It takes 2∼3 days training for 20 boxes cases. The Pytorch (Paszke et al., 2017) implementation of the CQL model is also open sourced.

## 5.3 Results

We evaluate the model on various number of packing boxes, specifically, 10 boxes, 16 boxes, 20 boxes and 30 boxes.

**Baselines:** The Multi-Task Selected learning (MTSL) (Duan et al., 2019) model, which initially designed for minimize the surface area of bin, and the authors adopt reinforcement learning for box selecting step, supervised learning for rotation step and a greedy method that minimizes bin surface usage for position step. For comparison purpose, we implement their model but set the reward function same as ours, specifically, the delta gap reward 5, and set the positioning step model similar to their rotation step model, which is the attention from encoder and previous decoder steps. To verify the effectiveness of our conditional query mechanism, we remove the conditional query of our model and get the box rotation and position from the box selection decoder. Besides, we also test the rollout baseline with REINFORCE (Williams, 1992) algorithm proposed on Kool et al. (2019), which they claimed it is more computationally efficient. Furthermore, the Genetic Algorithm (GA) proposed on Wu et al. (2010) is tested in 3D BPP.

**Metrics:** We evaluate previous mentioned algorithms by the bin gap ratio $r = 1 - \frac{\sum_{i=1}^{n} w_i l_i h_i}{W L H_n}$, which is positively related to the final bin height. The variance of bin gap ratio is also evaluated to show the stability of learning algorithms. It is worth noting that the optimal gap ratio of is greater than $0\%$. Depending on the dataset and box number, the optimal gap ratio is different. In general, the more boxes are required to pack, the more likely gaps are filled, and filling the gaps in 3D condition is even more difficult.

|    | Model | BIN10 | | BIN16 | | BIN20 | | BIN30 | |
|----|-------|-------|-----|-------|-----|-------|-----|-------|-----|
|    |       | Avg | Var | Avg | Var | Avg | Var | Avg | Var |
| 2D | Rollout | 38.8% | 0.0069 | 35.5% | 0.0056 | 30.5% | 0.0036 | 32.3% | 0.0028 |
|    | No query | 40.6% | 0.0077 | 29.3% | 0.0024 | 31.6% | 0.0037 | 28.3% | 0.0015 |
|    | CQL | **16.7%** | 0.0030 | **15.7%** | 0.0017 | **15.4%** | 0.0010 | **14.3%** | 0.0007 |
| 3D | MTSL | 50.2% | 0.0097 | 45.3% | 0.0065 | 42.6% | 0.0054 | 40.6% | 0.0037 |
|    | Rollout | 32.9% | 0.0158 | 43.2% | 0.0074 | 43.6% | 0.0067 | 49.0% | 0.0040 |
|    | No query | 46.5% | 0.0140 | 41.2% | 0.0069 | 38.3% | 0.0051 | 36.3% | 0.0032 |
|    | GA | **27.2%** | 0.0056 | 31.5% | 0.0030 | 33.9% | 0.0030 | 37.4% | 0.0023 |
|    | CQL | 27.4% | 0.0126 | **30.6%** | 0.0087 | **32.4%** | 0.0046 | **30.2%** | 0.0028 |

Table 2: The gap ratio of conditional query learning (CQL) and baselines

Table 2 shows the bin gap ratios and its variances on 512 test instances after 100 epochs training for learning algorithm. We use the default settings for genetic algorithm. The results show that our CQL model achieves low bin gap ratio and variance in most cases. It is clear that the model with conditional query mechanism is better than no query model, which justifies that the CQL fills the gap between sub-actions and makes the learning algorithm has the ability to reason following sub-action according the embedded of previous outputs. Meanwhile, The rollout baseline with REINFORCE produces similar results as no query model but with higher variance. Because the rollout method

sums up all reward signal in every packing steps as the action value, its learning process treat every packing step equally and back propagate gradients for all steps no matter the step is good or bad.

The overall gap ratio and variance of CQL decrease as the packing box number increases, which shows that the CQL model is scalable to problem size. Notice that in 10 boxes case of 3DBPP, the genetic algorithm achieve a little bit lower gap ratio than the CQL model, we analyze the results and find that the optimal solution of small packing box number tend to tile the bottom of bin, which is easy for heuristic algorithm to find good solutions. But when the box number goes larger, heuristic algorithms tends to drop into local optimum since they are more greedy and difficult to generalize the entire solution space. Besides, the genetic algorithm has relative lower variance compare with learning approaches because it use some fixed rules that result in similar solutions for different examples, which leads to low performance for specific instances. The end to end MTSL learning process shows poor results, because the model is not only partially observable MDP, but also suffers from sparse rewards due to the utilization of rollout method.

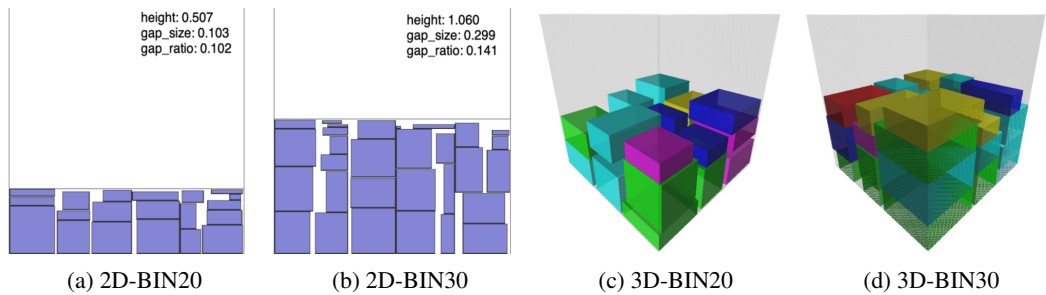

| (a) 2D-BIN20 | (b) 2D-BIN30 | (c) 3D-BIN20 | (d) 3D-BIN30 |

Figure 4: The Packing Results of CQL. It is worth to note that the goal is to minimize the final bin height. Whether the boxes are close to each other does not affect the final height in many cases, so the results do not show very neat arrangement but already achieve near optimal bin height.

From the learning carve of training process shown in Fig. 3, the CQL model also shows superior stability than other learning algorithm, which leads to lower variance as shown in Table 2. The learning carve of no query model is oscillating since the sampled result of earlier step do not pass to the later one. That is, the model can only estimate the solution that is overall good but not fit the specific example. The learning carve of rollout method shows a big jumping up in the beginning of training process because of the sudden baseline update after the first epoch. In contrast, the CQL model benefits from conditional query to construct a strict MDP and has meaningful reward signals from every box packing step, so it shows smooth convergence process.

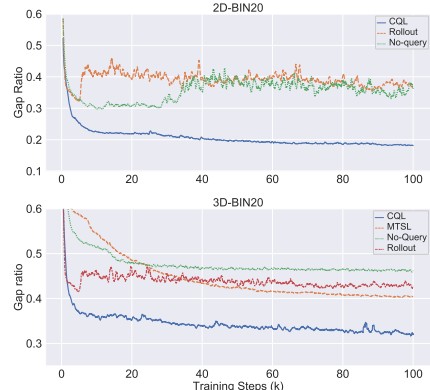

Figure 3: Learning carve of different models.

## 6 CONCLUSION AND FUTURE WORKS

In this paper, we propose the conditional query learning (CQL) model to solve packing problems. Benefit from the conditional query mechanism, the CQL model is capable of generalizing the problem that has mutual conditioned actions with relatively simple structure. Numerical results show that the conditional query learning greatly reduces the bin gap ratio both in 2D and 3D packing settings.

We are excited about the future of the conditional query model and we can apply it to other multi-dimension and parameterized action space problems. We can also extend the model with dynamic attention span and continuous action space to further improve its scalability. In addition, the current test environment does not consider the physical gravity of boxes and the boxes are not packed close to each other, we will add these restrictions to make it suitable for more practical bin packing.

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

# 7 APPENDIX

## 7.1 PACKING PROBLEM

Now we formally define the packing problem with notations in Wu et al. (2010).

$(l_i, w_i, h_i)$: parameters indicating the length, width, and height of box i.

$(L, W, \widetilde{H})$: length, width and height of the bin to be loaded, where $\widetilde{H}$ indicates that the bin height can be adjusted.

$(x_i, y_i, z_i)$: coordinates of box $i$'s left-bottom-behind corner.

$X_{l_i}, Z_{l_i}, Y_{w_i}, Z_{h_i}$: binary variables indicating the orientation of box $i$, where $X_{l_i}, Z_{l_i}$ indicate whether the length direction of box $i$ is parallel to the bin's $X$ and $Z$ axes, and $Y_{w_i}, Z_{h_i}$ defines the width direction is parallel to the $Y$ axis, or the height direction is parallel to the $Z$ axis respectively. For 3D view, there are six kind of orientation for a box.

$a_{ij}, b_{i,j}, c_{i,j}$: binary variables defining the relative placement of box $i$ to carton $j$: variables will be 1 if box $i$ is in front of, to the right of, or on top of box $j$, respectively; otherwise, 0.

$M$: a large enough number.

The objective is to minimize the variable bin height $\widetilde{H}$, that is $\min \widetilde{H}$

subject to the following set of constraints:

$$x_i + l_i X_{l_i} + w_i(Z_{l_i} - Y_{w_i} + Z_{h_i}) + h_i(1 - X_{l_i} - Z_{l_i} + Y_{w_i} - Z_{h_i})$$
$$\leq x_j + M(1 - a_{ij}), \qquad i \neq j \quad \text{(8a)}$$
$$y_i + w_i Y_{w_i} + l_i(1 - X_{l_i} - Z_{l_i}) + h_i(X_{l_i} + Z_{l_i} - Y_{w_i}) \leq y_j + M(1 - b_{ij}), \qquad i \neq j \quad \text{(8b)}$$
$$z_i + h_i Z_{h_i} + w_i(1 - Z_{l_i} - Z_{h_i}) + l_i Z_{l_i} \leq z_j + M(1 - c_{ij}), \qquad i \neq j \quad \text{(8c)}$$

$$x_i + l_i X_{l_i} + w_i(Z_{l_i} - Y_{w_i} + Z_{h_i}) + h_i(1 - X_{l_i} - Z_{l_i} + Y_{w_i} - Z_{h_i}) \leq L \qquad \text{(9a)}$$
$$y_i + w_i Y_{w_i} + l_i(1 - X_{l_i} - Z_{l_i}) + h_i(X_{l_i} + Z_{l_i} - Y_{w_i}) \leq W \qquad \text{(9b)}$$
$$z_i + h_i Z_{h_i} + w_i(1 - Z_{l_i} - Z_{h_i}) + l_i Z_{l_i} \leq \widetilde{H} \qquad \text{(9c)}$$

$$a_{ij} + a_{ji} + b_{ij} + b_{ji} + c_{ij} + c_{ji} >= 1, \qquad i \neq j \qquad \text{(10)}$$

$$X_{l_i} + Z_{l_i} <= 1 \qquad \text{(11a)}$$
$$Z_{l_i} + Z_{h_i} <= 1 \qquad \text{(11b)}$$
$$Z_{l_i} - Y_{w_i} = Z_{h_i} <= 1 \qquad \text{(11c)}$$
$$Z_{l_i} - Y_{w_i} + Z_{h_i} >= 0 \qquad \text{(11d)}$$
$$1 - X_{l_i} - Z_{l_i} + Y_{w_i} - Z_{h_i} <= 1 \qquad \text{(11e)}$$
$$1 - X_{l_i} - Z_{l_i} + Y_{w_i} - Z_{h_i} >= 0 \qquad \text{(11f)}$$
$$X_{l_i} + Z_{l_i} - Y_{w_i} <= 1 \qquad \text{(11g)}$$
$$X_{l_i} + Z_{l_i} - Y_{w_i} >= 0 \qquad \text{(11h)}$$

Constraints 8 ensure that any two boxes $i$ and $j$ do not overlap with each other. Constraints 9 keep all boxes with the bin dimension. $X_{l_i}, Z_{l_i}, Y_{w_i}$ and $Z_{h_i}$ are used to calculate the respective mappings of box length, width and height to the corresponding bin's $X, Y$ and $Z$ axes. Constraints 10 limits the relative position of any two boxes $i$ and $j$. Constraints 11 ensure that binary variables which determine the box position are properly controlled to reflect practical positions.

## 7.2 PACKING PROBLEM ENVIRONMENT DETAIL

In our packing problem environment, the model only needs to generate index, rotation and the horizontal coordinate of packing box in every packing step. That means that the environment automatically drops the box to the lowest available position in the bin.

$$z_i = max(z_b + h_b) \tag{12}$$

where box b satisfies:

$$
\begin{aligned}
x_i &< x_b + w_b \\
x_b &> x_i + w_i \\
y_i &< y_b + l_b \\
y_b &> y_i + l_i
\end{aligned} \tag{13}
$$

To avoid the model generates the positions outside of the bin, the environment forces the cross-border boxes to bin border.

$$
\begin{aligned}
x_i &= min(x_i, W - w_i) \\
y_i &= min(y_i, L - y_i)
\end{aligned} \tag{14}
$$

## 7.3 EXTEND RESULTS

Here, we illustrate the experiment results of 2D, 3D bin packing problem with different box numbers by showing the placement of boxes of different algorithms. Fig. 5 and Fig. 6 illustrate the results of 2D and 3D of CQL model in various packing box numbers. Obviously, when the number of the boxes increases, the boxes are placed denser and the gap ratio is become lower. Fig. 7 and Fig. 8 show the results of different algorithms of 2D & 3D BPP in 20 boxes case. It is clear that our algorithm performs better than others.

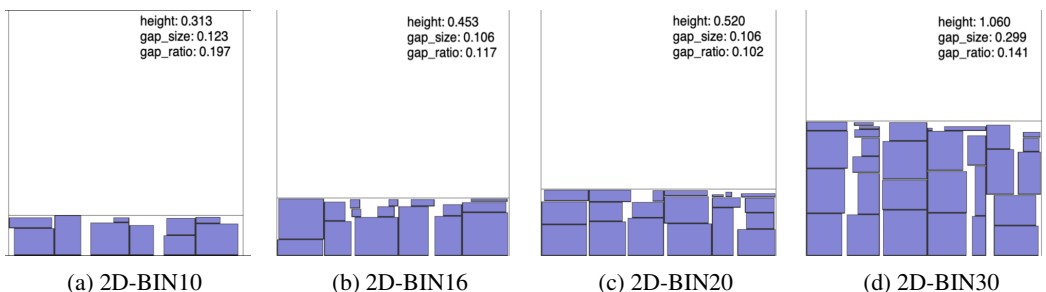

    (a) 2D-BIN10          (b) 2D-BIN16          (c) 2D-BIN20          (d) 2D-BIN30

Figure 5: The Packing Results of CQL in different cases of 2D BPP.

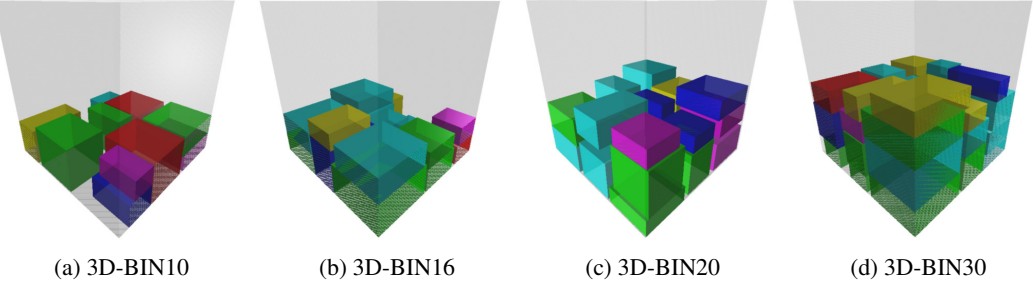

    (a) 3D-BIN10          (b) 3D-BIN16          (c) 3D-BIN20          (d) 3D-BIN30

Figure 6: The Packing Results of CQL in different cases of 3D BPP.

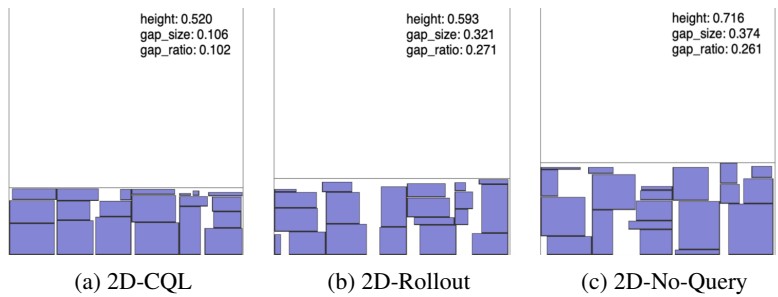

(a) 2D-CQL     (b) 2D-Rollout     (c) 2D-No-Query

Figure 7: The Packing Results of 2D BPP with different algorithms in the case of 20 boxes.

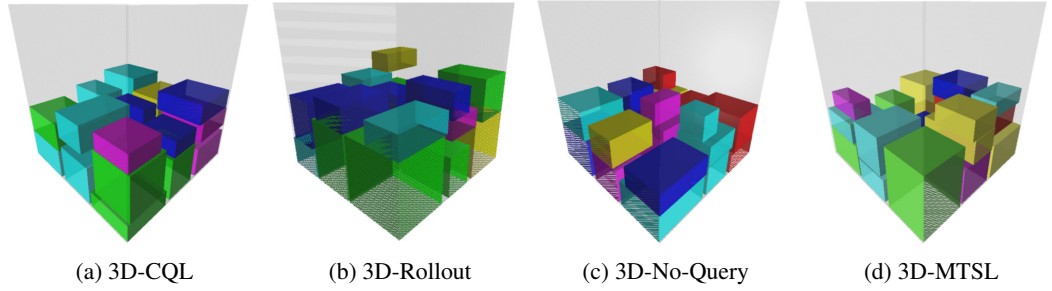

(a) 3D-CQL     (b) 3D-Rollout     (c) 3D-No-Query     (d) 3D-MTSL

Figure 8: The Packing Results of 3D BPP with different algorithms in the case of 20 boxes.

