# OpenReview forum: "Solving Packing Problems by Conditional Query Learning"
_ICLR.cc/2020/Conference — Reject_

### Official Review · AnonReviewer2 · 2019-10-08
**Official Blind Review #2**

**Rating:** 3

**Review:**

This paper proposes an end-to-end deep reinforcement learning-based algorithm for the 2D and 3D bin packing problems. Its main contribution is conditional query learning (CQL) which allows effective decision over mutually conditioned action spaces through policy expressed as a sequence of conditional distributions. Efficient neural architectures for modeling of such a policy is proposed. Experiments validate the effectiveness of the algorithm through comparisons with genetic algorithm and vanilla RL baselines.

Overall, the paper provides a solid contribution by proposing a new RL-based algorithm for the bin packing problem. In particular, the dense reward design for the MDP is quite interesting. I also think solving the 2D and 3D bin packing problem via RL is already quite valuable as an application.

However, I do not think the proposed method is novel enough. Most importantly, I perceive the concept of conditional query learning indifferent from the autoregressive modeling of the policy (with conditionally masked inputs). Although the authors divide the processing of each bin into three steps of MDP (not strictly), the steps can be merged into one without any change in the environment. Since other parts of the algorithm also come from the existing literature (except the reward design), I think this point is crucial for the paper to be published at the conference.

Minor comment:
- I think the genetic algorithm described in the experiments should be equipped with more details, i.e., the type of processor used and elapsed time.
- In table 2, I think bold numbers for the lowest variance of performance is very confusing since it does not necessarily mean a better algorithm.

**Experience Assessment:**

I have read many papers in this area.

**Review Assessment: Checking Correctness Of Derivations And Theory:**

N/A

**Review Assessment: Checking Correctness Of Experiments:**

I assessed the sensibility of the experiments.

**Review Assessment: Thoroughness In Paper Reading:**

I read the paper at least twice and used my best judgement in assessing the paper.

---

> ### Author Response · Authors · 2019-11-11
> **Response to Review #2**
>
> Thank you for your constructive comments and it seems you have understood our paper quite well. However, we want to reiterate our major contributions again.
>
> The conditional query model learning is a new method to address mutually conditioned action space problems.
>
> If sub-action steps are merged to one, the subsequent step can not condition on previous results, although the underline inference of results are saved in the network parameters, the specific actions are sampled from the probability distribution of model's outputs, so the merged model may perform overall not too bad. However, it can not produce a well fitted result for individual instance since the model lack of the knowledge of sampled results. By conditional query, the model get attention of the packing object over all packed boxes to inference next sub-action without effected by random sampling of previous actions.
>
> We conduct the experiment of merged model in the 'no query' model, the results shows that our conditional model greatly improves the performance of algorithm.
>
> The other two minor comments are very helpful, and we will revise it in our updated version.

---

### Official Review · AnonReviewer3 · 2019-10-20
**Official Blind Review #3**

**Rating:** 1

**Review:**

This paper aims at solving geometric bin packing (2D or 3D) problems using a deep reinforcement learning framework. Namely, the framework is based on the actor-critic paradigm, and uses a conditional query learning model for performing composite actions (selections, rotations) in geometric bin packing. Experiments are performed on several instances of 2D-BPP and 3D-BPP,

Overall, bin packing problems are challenging tasks for DRL, and I would encourage the authors to pursue this research topic. Unfortunately, I believe that the current manuscript is at a too early stage for being accepted at ICLR, due to the following reasons:

(a) The paper is littered with spelling/grammar mistakes (just take the second sentence: “With the developing” -> “development”). For the next versions of the manuscript, I would recommend using a spell/grammar checker.

(b) In the related work section, very little is said about Bin Packing Problems. There are various classes of BPPs, and it would be relevant to briefly present them. Moreover, BPPs have been extensively studied in theoretical computer science, with various approximation results. Again, a brief discussion about those results would be relevant. Notably, several classes of geometric bin packing problems admit polynomial-time approximation algorithms (for extended surveys about this topic, see e.g.  Arindam Khan’s Ph.D. thesis 2015; Christensen et. al. Computer Science Review 2017).

(c) According to the problem formulation and the experiments, it seems that the authors are studying a restricted subclass of 2D/3D bin packing problems: there is only “one” bin, so (it seems that) the authors are dealing with geometric knapsack problems (with rotations). Note that the 2D Knapsack problem with rotations admits a 3/2 + \epsilon - approximation algorithm (Galvez et. al., FOCS 2017). A. Khan has also found approximation algorithms for the 3D Knapsack problem with rotations. So, even if those results do not preclude the use of sophisticated DRL techniques for solving geometric knapsack problems, it would be legitimate to empirically compare these techniques with the polytime asymptotic approximation algorithms already found in the literature.

(d) The problem formulation is very unclear. Namely, the state representation is ambiguous: $s_p$ is obviously not a boolean variable, but a boolean vector (where each component is associated with an item). Nothing is said about actions and transitions and rewards (we have to read the AC framework in order to get a clue of these components). We don’t know if it is an episodic MDP (which is usually the case in DRL approaches to combinatorial optimization tasks). Also, it seems that the MDP is specified for a single instance of 3D-BPP. But this looks wrong since it should include the distribution of all instances of 3D-BPP.

(e) The Actor-Critic framework, coupled with a conditional query learning algorithm, is unfortunately unintelligible due to the fact that many notations are left unspecified. For example, in Eq (1) what are the dimensions K and V? In Eq (2) what is d_i? In the algorithm what is n_{gae}? Also in the algorithm, what are l’_i, w’_i and h’_i? Etc.

(f) Even if the aforementioned issues are fixed, it seems that the framework is using many hyper-parameters (\gamma, \beta, \alpha_t, etc.) which are left unspecified. Under such circumstances, it is quite impossible to reproduce experiments.


**Experience Assessment:**

I have read many papers in this area.

**Review Assessment: Checking Correctness Of Derivations And Theory:**

I assessed the sensibility of the derivations and theory.

**Review Assessment: Checking Correctness Of Experiments:**

I assessed the sensibility of the experiments.

**Review Assessment: Thoroughness In Paper Reading:**

I read the paper at least twice and used my best judgement in assessing the paper.

---

> ### Author Response · Authors · 2019-11-11
> **Response to Review #3**
>
> Thank you for your constructive comments and the acknowledgement of our research direction. we address your  concerns below:
>
> a) Indeed, there are a few typos and grammar mistakes, and we will fix that carefully in the new version.
>
> b) In the related works section, we do not introduce which class of BPPs we try to solve, actually we describe this in the preliminaries section and put the the formal definition of the problem in Appendix 7.1 since the page limitation of  main paper. Actually the problem we are solving is the strip packing problem with rotations. And for theoretical studies on traditional BPPs, yes, there are extensive traditional studies on BPPs as we mentioned in the introduction. But in this paper, we actually propose a new conditional query mechanism to solve the problems which has several mutual conditioned actions like BPPs, so we do not put much words on the traditional studies on BPPs. And for the BPPs,  approximation algorithms are somewhat relevant and we will discuss this briefly in the revision.
>
> c) Here we try to address that the our new model is much better than previous Neural Combinatorial Optimization (NCO) methods, so we compare our results with some recent NCO solutions (Duan et al., 2019, Kool et al. 2019) to show the effectiveness of our model.
>
> d) In the problem formulation, actually the $s=(s_p,l_i,w_i,h_i,x_i,y_i,z_i)$ is the state of one box, so the $s_p$ is a boolean variable, and the state of the whole problem instance is the combination of all box states, and we will make this clearer later.
>
> We describe the action in the section 3.1 and divide the action to sub-actions for the conditional query model, and we also recall it in the section 4, and section 5.2. The reward function is formulated bin section 4.3.3. Our MDP is an episodic MDP, it is obvious that the packing problem has the terminal state, In reinforcement learning, the goal is to maximize accumulated reward of a episode, this also mentioned in the reward function designing part, and we will make this clearer later.
>
> The MDP is not specified for a single instance of 3D-BPP. Our model has the ability to generalize for the various instances, In our section 5.1, we describe that our dataset is uniformly sampled to form different instances.
>
> e) We will make the notations of formulations clearer later, the dimensions of K and V is hyper-parameter of the attention model (ref. Vaswani et al., 2017). We set it to 64 as described in section 5.2. In Eq(2) d_i is the dimension of key (ref. Vaswani et al., 2017), and we will clarify this in Appendix. In Algorithm, $n_{gae}$ is the rollout steps of GAE, and $l'_i,w'_i,h'_i$ are the rotated box size of current packing box, which we defined in section 3.2 and Table 1, we will recall this definition in the algorithm part.
>
> f) We think that using hyper-parameters is not a problem, every reinforcement learning algorithm has the hyper-parameter $\gamma$ as discount ratio, and in our setting, we just set it as 1 since we not want a greedy algorithm (ref. section 4.3.3). And we will add a description of hyper-parameters.
>
> For reproduction of our algorithm, the ICLR encourage all authors open-source their code and provide a source code link when submitting their papers. As we mentioned in abstract, we released our model and test environment, those interested in this work can just download the code and run it to reproduce our experiments.

---

### Official Review · AnonReviewer1 · 2019-10-26
**Official Blind Review #1**

**Rating:** 6

**Review:**

Summary:
The paper proposes heuristics to solve the bin packing problems based on reinforcement learning with deep neural networks. With a new heuristics of conditional queries, the proposed method works favorably with the previous RL-based approach and other baselines.


Comments:

The idea of applying reinforcement learning to combinatorial optimization itself is not new. The authors, on the other hand, propose new heuristics, called conditional queries, which divides a unit of actions (rotation, box, and etc.), which turns out to be effective compared to the previous reinforcement-learning based method.




**Experience Assessment:**

I have published in this field for several years.

**Review Assessment: Checking Correctness Of Derivations And Theory:**

I assessed the sensibility of the derivations and theory.

**Review Assessment: Checking Correctness Of Experiments:**

I assessed the sensibility of the experiments.

**Review Assessment: Thoroughness In Paper Reading:**

I read the paper at least twice and used my best judgement in assessing the paper.

---

> ### Author Response · Authors · 2019-11-11
> **Response to Review #1**
>
> Thank you for taking the time to review our paper and for your positive feedback.

---

### Decision · Program_Chairs · 2019-12-19

**Decision:**

Reject

**Comment:**

This paper proposes an end-to-end deep reinforcement learning-based algorithm for the 2D and 3D bin packing problems. Its main contribution is conditional query learning (CQL) which allows effective decision over mutually conditioned action spaces through policy expressed as a sequence of conditional distributions. Efficient neural architectures for modeling of such a policy is proposed. Experiments validate the effectiveness of the algorithm through comparisons with genetic algorithm and vanilla RL baselines.

The presentation is clear and the results are interesting, but the novelty seems insufficient for ICLR. The proposed model is based on transformer with the following changes:
* encoder: position embedding is removed, state embedding is added to the multi-head attention layer and feed forward layer of the original transformer encoder;
* decoder: three decoders one for the three steps, namely selection, rotation and location.
* training: actor-critic algorithm